# Revolutionizing the Early Detection of Alzheimer’s Disease through Non-Invasive Biomarkers: The Role of Artificial Intelligence and Deep Learning

**DOI:** 10.3390/s23094184

**Published:** 2023-04-22

**Authors:** Aristidis G. Vrahatis, Konstantina Skolariki, Marios G. Krokidis, Konstantinos Lazaros, Themis P. Exarchos, Panagiotis Vlamos

**Affiliations:** Bioinformatics and Human Electrophysiology Laboratory, Department of Informatics, Ionian University, 49100 Corfu, Greece

**Keywords:** Alzheimer’s disease, early prediction, non-invasive biomarkers, explainable AI, deep learning

## Abstract

Alzheimer’s disease (AD) is now classified as a silent pandemic due to concerning current statistics and future predictions. Despite this, no effective treatment or accurate diagnosis currently exists. The negative impacts of invasive techniques and the failure of clinical trials have prompted a shift in research towards non-invasive treatments. In light of this, there is a growing need for early detection of AD through non-invasive approaches. The abundance of data generated by non-invasive techniques such as blood component monitoring, imaging, wearable sensors, and bio-sensors not only offers a platform for more accurate and reliable bio-marker developments but also significantly reduces patient pain, psychological impact, risk of complications, and cost. Nevertheless, there are challenges concerning the computational analysis of the large quantities of data generated, which can provide crucial information for the early diagnosis of AD. Hence, the integration of artificial intelligence and deep learning is critical to addressing these challenges. This work attempts to examine some of the facts and the current situation of these approaches to AD diagnosis by leveraging the potential of these tools and utilizing the vast amount of non-invasive data in order to revolutionize the early detection of AD according to the principles of a new non-invasive medicine era.

## 1. Shedding Light on Alzheimer’s Disease: The Current Situation and What We Know So Far

The current statistics on Alzheimer’s disease (AD) are alarming and paint a grim picture of the future. AD is a widespread and prevalent disease across the globe and is the leading cause of dementia in individuals aged 65 and above [1]. According to reports from the UN Aging Program and the US Centers for Disease Control and Prevention, it is projected that the global population of individuals aged 65 and above will increase from 420 million in 2000 to almost 1 billion by 2030 [2]. This demographic shift will result in a substantial increase in the proportion of older individuals, rising from 7% to 12% of the world’s population. According to the Special Report examination, an estimated 6.7 million people aged 65 and older are living in the USA with AD, and this number could be increased to 13.8 million by 2060 [3].

Meanwhile, AD is a complex and multifactorial disorder that arises from a combination of genetic, environmental, epigenetic, and metabolic factors [4]. Alongside brain-related etiopathogenic mechanisms, these factors contribute to the heterogeneous cognitive phenotype that characterizes AD. This complexity has posed significant challenges to the development of effective drug treatments for AD, as evidenced by the high rate of failure of drug candidates entering the AD drug-development pipeline over the past three decades [5]. 

Despite significant research and development in the past few decades to create new diagnostic algorithms that utilize biomarkers , the diagnostic rate for Alzheimer’s disease (AD) remains low [6]. The hallmark of AD is characterized by specific cerebrospinal fluid (CSF) biomarkers , such as the accumulation of amyloid beta peptide (Aβ) which can be accurately detected in CSF samples of AD patients. Furthermore, a recent cross-sectional analysis investigated the inter-play between A*b* and tau in early AD and its impact on cognitive impairment [7]. The study utilized CSF markers and found that cognitive and memory performance were significantly associated with tau levels in the early stages of AD, with the correlation being dependent on A*b*. A recent indicative study [8] highlighted the potential of assessing CSF molecules in distinguishing between AD and non-AD patients. Their findings revealed that the ratio of A*b*42 to A*b*40 exhibited the most effective discriminatory ability and may be valuable in clinical practice. As a result, there is a huge amount of literature focused on the use of CSF-based approaches for the early diagnosis of AD. 

While the use of CSF biomarkers for AD diagnosis shows promising results, it is important to note that it is an invasive technique with potential drawbacks. The procedure requires a lumbar puncture, which can be uncomfortable and carries a small risk of side effects [9]. Additionally, the cost and availability of the test may limit its widespread use. It should also be noted that even minor bio-chemical changes in the brain can be reflected in the composition of the CSF [10]. Consequently, CSF is widely considered as a high-quality biological fluid that has the potential to contain validated clinical signatures for AD; however, it is important to recognize its limitations and drawbacks.

In the present work, in Section 2, we discuss the need for early diagnosis of AD from a non-invasive perspective. Section 3 explores cutting-edge studies on the early onset of the disease using non-invasive techniques, while in Section 4, we present recent advances in non-invasive AD diagnosis using deep learning and machine learning techniques. Finally, in Section 5, we discuss the role of artificial intelligence and deep learning as novel insights into AD monitoring. This study emphasizes the importance of non-invasive methodologies for the early diagnosis and monitoring of AD, with a particular focus on the potential of artificial intelligence methods for complex data analysis.

## 2. Need for Early Diagnosis of AD—The Non-Invasive Perspective 

It is widely acknowledged in the scientific community that AD is a debilitating and progressive neurodegenerative disorder with no known cure. As such, the current focus of AD research is shifting towards early detection and intervention, as this is considered critical to improving patient outcomes and developing more effective therapies. The pre-symptomatic stage of AD, during which cognitive decline is yet to manifest clinically, has emerged as a key window of opportunity for early detection and intervention [11]. Currently available therapeutics for the management of AD offer solely symptomatic relief. Over the last ten years, numerous compounds with disease-modifying potential have been brought to clinical trials and proven ineffective. The limited success of molecular therapeutic approaches has led to an inclination towards non-invasive therapies as an alternative, which can circumvent most of the challenges encountered by their molecular counterparts [12]. 

By identifying biomarkers of AD pathology in the pre-symptomatic phase, it may be possible to predict the onset of cognitive symptoms and enable the implementation of preventative measures and targeted interventions. The importance of early detection and intervention in AD will lead to the development of more personalized treatment strategies, incorporating both pharmacological and non-pharmacological interventions, that can better meet the complex needs of AD patients. Ultimately, this will require ongoing research efforts and collaborations across multiple disciplines to develop a more comprehensive understanding of AD pathology and effective interventions.

The urgent need to implement effective diagnostic methods that can detect the disease at an early stage is inextricably linked to the use of non-invasive methods. Such approaches for early detection of AD have gained considerable attention in recent years, owing to their accessibility, safety, and cost-effectiveness. A non-invasive perspective on the challenges of AD is an important area of research that has the potential to revolutionize the way the disease is diagnosed and managed. Non-invasive techniques for the early prediction of AD include a range of methods. Imaging techniques, such as magnetic resonance imaging (MRI) and positron emission tomography (PET) [13], can provide insights into the structural and functional changes in the brain associated with AD symptoms. These techniques can detect early signs of neurodegeneration, such as changes in brain volume, reduced glucose metabolism, and the accumulation of beta-amyloid plaques. Furthermore, analysis of blood-based samples can also provide early indications of AD pathology, by measuring levels of AD-related proteins such as Aβ and tau [14]. In addition to these established techniques, digital data from wearables and bio-sensors are emerging as promising non-invasive tools for early prediction of AD. Wearable devices, such as smartwatches and fitness trackers, can collect data on physical activity, sleep patterns, and other lifestyle factors that may be linked to AD risk, while bio-sensors can monitor changes in physiological parameters, such as heart rate and breathing, which may reflect early signs of AD pathology [15]. Tarnanas and Rapp, introduced a novel digital bio-marker, the neuro-motor-index (NMI), that can provide personalized prognostic information and feasible scalability [16]. NMI was proven to detect subtle changes in pre-clinical AD patients starting 24 months before any symptoms of cognitive impairment develop. There is a variety of other sensor-based approaches and applications that can be used for distinguishing AD, as described below.

## 3. Cutting-Edge Studies for Early-Onset AD through Non-Invasive Techniques

### 3.1. Wearable Devices for Digital Biomarkers

There is a growing body of literature suggesting that cognitive, behavioral, and motor changes may occur years prior to the onset of clinical symptoms in individuals with AD [17]. As researchers strive to establish a gold standard for the assessment of the disease, there is a growing interest in the identification of easily accessible digital biomarkers that utilize advancements in consumer-grade mobile and wearable technologies. In recent years, a notable advancement has been made in the identification of digital biomarkers for early prediction of AD through the analysis of wearable data [18]. Indicatively, this study provided evidence that digital bio-marker prognostic models can serve as a valuable tool to aid in the large-scale screening of populations for early detection of cognitive impairment, as well as for ongoing patient monitoring [18]. The authors utilized a smartphone/tablet-based digital bio-marker that implemented augmented reality tasks inspired by complex functional instrumental activities of daily living (Altoida iADL tasks). These tasks primarily focused on assessing spatial memory and navigation abilities. Along the same lines, Lancaster et al. [19] introduced the Gallery Game, an episodic memory task that serves as a digital bio-marker for Alzheimer’s disease. This bio-marker has demonstrated a significant ability to identify individuals in the pre-clinical stages of the disease, thereby enabling more effective recruitment for clinical trials and enhancing the tracking of disease progression and treatment response.

### 3.2. Sensors-Based Biomarkers

Data obtained from sensors have the potential to enable early prediction. By collecting digital biomarkers related to eye movements, pupillary reflexes, speech, and other relevant features, sensors can be employed to detect changes in these biomarkers that may be indicative of the early stages of AD. Indicatively, the acquisition of digital biomarkers related to eye movements and pupillary reflexes is commonly accomplished through the utilization of cameras and light sensors. These sensors have significant potential for employment in AD since individuals with Alzheimer’s disease may demonstrate early-stage deficits in eye movements [20]. 

Early detection of biomarkers associated with diseases is of utmost importance for the development and implementation of effective treatments, which can alleviate the disease’s progression and enhance the patient’s quality of life. In their research, Haider et al. (2020) [21] utilized an open-source software suite, namely the openSMILE v2.1 toolkit, to evaluate acoustic features for the detection and diagnosis of AD. By automatically extracting speech features and applying machine learning methods, this toolkit produced reasonably accurate outcomes for AD detection. Thus, the use of this toolkit could potentially serve as a cost-effective and non-invasive screening method for AD in comparison to blood or imaging biomarkers . 

### 3.3. Blood-Based Biomarkers for AD-Related Brain Changes

Brain-derived biomarkers are typically detected at relatively low concentrations compared to cerebrospinal fluid because of the blood-brain barrier, which prevents the free passage of substances between the central nervous system and circulation compartments. However, blood analysis, and mainly non-invasive plasma analysis, presents advantages as a promising screening tool, and improvements have been developed to increase analytical sensitivity. Moreover, combining blood bio-marker detection with imaging markers may help enhance the accuracy of AD diagnosis [22]. The measurement of high-performance plasma amyloid-*b* biomarkers (APP)669-711/amyloid-*b* (A*b*)1-42 mass spectrometry (IP-MS) was performed, indicating high performance when predicting brain amyloid-*b* burden [23]. 

Amyloid *b* 1-42 (A*b*)1-42, total tau (t-tau), p-Tau-181, and neurofilament light chain (NFL) are potential biomarkers for AD. In a recent study, a plasma-based primary screening of NFL/(A*b*)1-42 was introduced as a strong bio-marker reflecting brain neurodegeneration and amyloid pathology in AD. Plasma NFL/(A*b*)1-42 was associated with high diagnostic accuracy. In early AD while in a pre-clinical state, it changed more rapidly than the compared CSF for tau (t-Tau or p-Tau-181), demonstrating this ratio as a non-invasive plasma-based bio-marker for early diagnosis and monitoring of AD progression [24]. 

Plasma tau phosphorylated, at threonine 181 (p-tau181) level quantification, was significantly enhanced in AD patients compared to the control cohort following an ultrasensitive Simoa immunoassay, demonstrating p-tau181 as a promising blood bio-marker for the detection of brain AD pathology [25]. Brain-derived tau as a new blood-based bio-marker that outperformed plasma total-tau indicated specificity to AD-type neurodegeneration. Total tau levels were also measured from stored plasma samples in Framingham Heart’s study participants, using single-molecule array technology, suggesting that this marker may improve the prediction of future AD dementia [26]. Longitudinal alterations in plasma p-tau181 and plasma neurofilament light chain levels were related to prospective neurodegeneration and cognitive decline and can support monitoring of disease progression in AD, as was provided by a recent study with participants from the multi-centric Alzheimer’s Disease Neuroimaging Initiative study [27].

### 3.4. Bio-Sensors for Biofluid Marker Detection in AD

Bio-sensors are analytical devices that convert biological reactions into measurable signals. Along with numerous approaches in clinical practice, such as enzyme-linked immunosorbent assays, immunohistochemistry, mass spectrometry, magnetic resonance imaging, and positron emission tomography, their contribution can bring many improvements to AD monitoring by detecting circulated biomarkers in bio-fluids with high sensitivity and specificity. A new chain-shaped electrode was developed to avoid the edge effect of the electric field distribution by immobilizing a specific anti-A*b* antibody onto a self-assembled monolayer functionalized inter-digitated chain-shaped electrode designed to improve the sensing area homogeneity. Recent studies highlight various types of bio-sensors, which include electrochemical, fluorescent, and (FET)-based sensor configurations for AD clinical monitoring [28].

The bio-sensor was characterized as highly sensitive for the non-Faradaic detection of (A*b*)1-42 peptide in human serum at different concentrations and bypassed the denaturation of the protein caused by the metallization between the protein and ferric ion in the redox probe [29]. In a similar work, the design of an A*b* electrochemical aptasensor was performed using a fern leaf-like gold nano-structure as a transducer, a specific RNA aptamer as a recognition element, and ferro/ferricyanide as a redox marker [30]. Amyloid-*b* oligomers (A*b*O) are important diagnostic markers for AD. Qin et al., prepared a hierarchical gold electrode with gold dendrites and dendritic electropolymerized poly(pyrrole-3-carboxylic) acid substrate for the immobilization of prion protein and the selective detection of A*b* oligomers. The sensor offers high conductivity, high selectivity, and a large surface area and could be used for A*b* oligomer measurement in blood samples of AD patients [31].

A label-free electrochemical bio-sensor has also been designed for the specific recognition of A*b*O based on the binding of these markers to specific thiol-terminated ssDNA aptamer receptors, attached to gold electrodes, which can be used for the early detection of AD symptoms [32]. A sensitive electrochemical immunosensor platform has been reported for the quantitative detection of A*b* peptides, using gold nano-particle functionalized chitosan-aligned carbon nano-tube nano-composites on glassy carbon electrodes. The sensor performance was evaluated in diluted serum and presented remarkable results against peptide detection, which may be helpful for diagnosis and disease management [33].

A reduced graphene oxide-based field-effect transistor bio-sensor for in situ analysis was designed against AD to monitor the enzymatic kinetics of acetylcholinesterase and acetylcholine, which could serve as a potential application in clinical practice as well as in the treatment of the disease [34]. A graphene oxide-based single-use electrochemical bio-sensor was developed for the sensitive and selective detection of serum miRNA-34a. PGEs were used for developing a disposable bio-sensor platform, graphene oxide was modified onto the pencil graphite electrode surface by EDC-NHS chemistry, and different experimental conditions such as DNA probe concentration, miRNA-34a target concentration, and hybridization time were performed during the development steps [35]. A cost-effective method for the clinical diagnosis of AD was introduced based on the electrochemical detection of clusterin in spiked plasma using a screen-printed carbon electrode modified with 1-pyrenebutyric acid N-hydroxysuccinimide ester and decorated with specific anti-clusterin antibody fragments [36].

Razzino et al. developed an amperometric immunosensor for the sensitive measurement of tau protein by implementing a sandwich immunoassay onto disposable screen-printed carbon electrodes grafted with p-aminobenzoic acid, modified with a gold nano-particle-poly(amidoamine) (PAMAM) dendrimer nano-composite and involving a horseradish peroxidase-labeled detector antibody [37]. A neutrally charged immunosensor was developed by Dai et al. and offers a simple, rapid fabrication method for routine clinical analysis of tau protein. The bio-sensing platform, based on the pico-molar level detection limit, could be sufficiently low for quantification of this characterized AD marker. [38]. A nanocomposite bio-sensing platform consisting of an indium tin-gold electrode coated by polyethylene terephthalate was constructed for tau-441 antibodies binding in serum using a nano-composite of reduced graphene oxide and gold nano-particles, which presented high efficiency for the immobilization of the antibody and selective determination of the protein [39]. A novel electrochemical aptamer-antibody sandwich assay for the detection of another tau isoform, tau-381, in human serum, was designed, combining the advantages of signal amplification of the gold nano-particles and increasing the affinity of the aptamer through cyclic voltammetry, electrochemical impedance spectroscopy, and differential pulse voltammetry techniques [40]. Similar approaches, including sensor-based platforms, have been implemented in other NDs, such as Parkinson’s disease, as recently presented [41].

### 3.5. PET and MRI Data for Imaging Biomarkers

Magnetic resonance imaging (MRI) and positron emission tomography (PET) are two medical imaging techniques that are widely used for the early detection of AD [42]. MRI is a completely non-invasive and safe medical imaging technique that does not require any invasive procedures, such as needles or injections. On the other hand, PET scans involve the injection of a radioactive tracer, but they can still be considered non-invasive since no cutting or instrument insertion into the body is necessary. The procedure of inserting the radioactive tracer is similar to getting a routine blood test, and it is usually painless, causing no discomfort. 

By combining medical imaging techniques such as MRI and PET with deep and machine learning methods, non-invasive and early diagnosis strategies for neurodegenerative diseases, such as Alzheimer’s can be developed. For example, Ding et al. [43] used prospective F-FDG PET brain scans to train a 48-layer deep convolutional neural network called InceptionV3. This model outperformed three radiology readers in the ROC space, suggesting it could be a promising non-invasive medical decision support system. Mehmood et al. [44] used MRI images with a pre-trained VGG convolutional neural network to achieve high accuracy in distinguishing between AD patients and normal controls. Their model also had high classification accuracy in distinguishing between early mild cognitive impairment (EMCI) and late mild cognitive impairment (LMCI). Guo et al. [45] proposed a graph-based convolutional neural network architecture called PETNet, which analyzes PET scans as signals on a group-wise inferred graph structure. By modeling PET images as signals on a network structure, PETNet offers an effective and computationally efficient approach for medical image analysis and the early diagnosis of Alzheimer’s compared to other machine learning models. 

### 3.6. Sensors for Oculomotor Functions

Cameras and light sensors are most commonly used to collect digital biomarkers linked to eye movements and pupillary reflexes. Such sensors can be used in disease diagnosis and overall research. People with AD can show impairments in eye movements very early in the disease [46]. Visuomotor network dysfunctions may be a potential bio-marker for Alzheimer’s disease (AD) as well as mild cognitive impairment (MCI), a prodromal stage of AD. The functionality of this network was tested using goal-directed eye-hand tasks recorded with a head-mounted video infrared eye-tracking system (Chronos and EyeSeeCam). The data collected suggests that dysfunctions of the visuomotor network could be used as biomarkers for research purposes. Additionally, the proposed eye-hand tasks could help to solidify and provide a clear definition of the pre-clinical phenotype of AD [47]. Another study also supported the use of eye-tracking tests (utilizing an eye-tracking device—Gazefinder NP-100, JVC KENWOOD Corporation, Kanagawa, Japan—to distinguish cognitive functions between controls, MCI, and AD patients [48]. Boz et al. showed in a recent study, which included a cohort of AD patients, amnestic mild cognitive impairment, and neurotypical adults, that correct saccade rates and latency may be distinguishing parameters of early AD [49]. Jonell et al., 2021, proposed a system that uses non-invasive tools to record data from multiple sources during clinical interviews as part of AD cognitive assessments. The system utilizes the following sensor devices: smartphones, tablets, eye trackers, a microphone array, and a wristband. The findings suggest that this data can be used to improve the clinical assessment of early dementia [50]. Bartoli et al., 2017, described a low-cost robotic interface that can be used to record oculomotor functions in AD patients [51]. In particular, it measures the hand trajectory, reaction times, and movement tracking errors. The Omni robot (Omni^®^, Sensable) is a tool that may be easily installed on or beside any desktop and employed in outpatient clinics. The findings corroborate the notion that memory capacity and use of visuo-spatial correlations may underpin motor behavior impairment observed in AD patients. NeuroGlasses, which are wearable sensor-based glasses, were also implemented in a cohort of early-stage AD patients to assess, for the first time, the number of blinks and the essential tremor of the head [52]. 

### 3.7. Sensors for Movement, Speech, and Language Functions

One of the symptoms of AD is the progressive decline of motor functions, as well as degeneration in cognitive functions. Early diagnosis ensures timely treatment and improved quality of life for the patient. A study by Serra-Añó et al., 2019, used smartphone sensors in order to assess the mobility of people with AD [53]. The software FallSkip^®^ (Biomechanical Institute of Valencia) was utilized via an Android device (Xiaomi Redmi 4x Model MAG138) to evaluate how well individuals with various stages of AD and those without dementia performed on a variety of tasks. Both mild and moderate AD patients exhibit impairments in some important motor functions, including gait, turning and sitting, standing from a sitting position, and reaction time [53]. Suzumura et al., 2018, used the JustTouch screen to record eight parameters in order to screen for finger function and detect finger dexterity irregularities in AD patients [54]. In conclusion, decreased finger dexterity can be a sign of deteriorating cognitive function. This smart tool that records finger dexterity can be used to facilitate the screening of MCI and AD. Alvarez et al. monitored the daily motion and detected abnormal behavior and gait abnormalities through motion location tracking and body signal processing [55]. A foot-mounted wearable sensor device including 9-axis inertial sensors (accelerometer, gyroscope, and magnetometer) named Sensor Foot was developed to collect walking data and explore the benefits of aerobic activity along with traditional cognitive protocols [56]. Lu et al. used a wrist-worn accelerometer for a short period of time to compare patterns of physical activity and sedentary behavior among AD and MCI patients and normal cognitive individuals, showing differences between patients and healthy cohorts [57].

### 3.8. Sensors for Autonomic Nervous System Functions

The autonomous nervous system (ANS) has been shown to undergo pathological changes in AD, according to some studies [58]. ANS disruptions can be an efficient way to detect AD at an early stage, seeing as autonomic brainstem nuclei are among the first areas of the brain affected by AD-related tau aggregation, years before the onset of cognitive symptoms [59]. A key marker of ANS balance is heart rate variability (HRV) [60]. Taking into account the connection between HRV and cognition, the early cholinergic and parasympathetic system disruption, in addition to its steady deterioration in AD, HRV can be considered a convincing diagnostic bio-marker for AD development. Heart rate sensors using photoplethysmography (PPG) technology have been proposed for monitoring heart rates in AD patients [61]. The degree of cognitive function has been demonstrated to be inversely connected with HRV in AD and MCI populations, where parasympathetic activity is inhibited as a result of damage to the cholinergic systems [62]. Gwak et al., 2019, suggested a novel approach to MCI diagnosis, comprising a commercial wristwatch, a wireless pulse oximeter, a photoplethysmography (PPG), and gait (accelerometer and gyroscope) sensors to identify predictors [63]. The sensor-derived data was used to distinguish between MCI and cognitively healthy subjects. The results of this study demonstrate the potential of sensor-derived measurements to support disease diagnosis and reduce the labor of healthcare professionals in this field. Table 1 summarizes non-invasive sensor-based approaches to AD monitoring.

## 4. Recent Advances in Non-Invasive AD Diagnosis Using DL and ML Techniques 

In this section, we provide an overview of the recent cutting-edge DL and ML methodologies for non-invasive AD data. One promising aspect of studies in this field is the use of wearable data, which provides a purely non-invasive technique for collecting digital data. This approach holds great potential for improving the accuracy and effectiveness of non-invasive AD diagnosis. Under this perspective, utilizing advanced ML-based classification models for early diagnosis, gait data was collected from 145 subjects using wearable sensors with built-in inertial measurement units [64]. The collected data was used to train an elimination method-based ensemble and oversampling model, which demonstrated high accuracy in detecting the early stages of AD. The focus on enhancing the machine learning aspects of the methodology highlights the potential of utilizing deep learning (DL) and machine learning techniques to improve the accuracy and efficiency of AD diagnosis. Ultimately, this can lead to improved patient outcomes. Similarly, a novel approach for early identification of AD using machine learning-based classification models that analyze spectrogram features extracted from speech data is presented [65]. In the experiments, the authors compared different machine learning methods using speech data from AD and healthy control (HC) subjects. They highlight the potential of utilizing machine learning-based classification models to analyze AD speech data, offering a promising approach to address the challenges of early diagnosis of AD. 

Significant progress has also been made in the field of imaging data analysis, particularly in the application of DL processes. In an indicative study [66], the potential of DL techniques was explored in the detection of AD and its core pathologies, including amyloid pathology, tau pathology, and neurodegeneration, using voxel-based analysis of structural MRI. The researchers implemented a three-dimensional convolutional neural network (3D CNN), trained with a data augmentation strategy to classify Alzheimer’s dementia, and generated class activation maps. This method offers practical advantages by potentially reducing patient burden, risk, and cost when extracting bio-marker information from conventional MRIs using DL techniques. Moreover, a hybrid model (called CNN-SVM) that combines CNN and the support vector machine (SVM) classifier to predict the early stages of AD from MRI data is proposed [67]. The results of the study show the potential of this model, highlighting the significance of using hybrid deep and machine learning techniques for the early detection of Alzheimer’s disease.

An effective computational method for the diagnosis of Alzheimer’s disease from brain MRI scans was recently proposed [68]. It involves two phases: segmentation and classification, both based on deep and machine learning. The segmentation model combines the Gaussian mixture model (GMM) and CNN to segment brain tissues, while the classification model combines extreme gradient boosting (XGBoost) and a support vector machine (SVM) to classify Alzheimer’s disease based on the segmented tissues. The authors conclude that DL techniques are effective for segmentation and feature extraction in medical image processing and that the combination of XGboost and SVM improves classification results.

Blood-based data is gaining ground as a promising AD non-invasive bio-marker. Such data are typically generated with omics techniques and exhibit high complexity, large data volumes, and high dimensionality, making it challenging to extract meaningful information manually. Therefore, the use of DL and machine learning processes has become increasingly important for analyzing and interpreting blood-based omics data. In this context, a recent study focuses on utilizing machine learning algorithms to discover small sets of blood transcripts that can be distinguished between healthy individuals and those with neuro-degenerative diseases, including Alzheimer’s disease and other neuro-degenerative diseases [69]. The researchers developed a tree-based machine learning algorithm and applied it to transcripts present in blood, resulting in the discovery of small sets of transcripts that can be used to distinguish between these diseases with high sensitivity and specificity. Furthermore, the study [70] focuses on combining molecular markers as “weak learners” using machine learning approaches to obtain more accurate diagnostic results for AD. Various machine learning approaches, including support vector machines, decision trees, neural networks, and gradient-boosted trees, were explored. 

A combination of DL models with imaging from blood samples to investigate AD is suggested [71]. According to this process, the potential link between AD and morphologically abnormal neutrophils on blood smears can be achieved. DL models were trained to predict AD from neutrophil images due to the complexity and subjectivity of the task by human analysis. Control models were also trained for a known feasible task and for detecting potential biases of overfitting. The authors concluded that a solid DL pipeline with positive and bias control models and visualization techniques is helpful to support DL model results.

An AI pipeline to evaluate the accuracy of putative cytosine epigenetic markers for Alzheimer’s disease (AD) diagnosis was recently published [72]. Methylation profiling of circulating cfDNA was collected from individuals suffering from AD and cognitively healthy controls. The study employed six AI algorithms, including DL, to perform classification and regression analysis. The study also highlighted the importance of the hierarchical learning process in DL and the calculation of the activation value of the hidden layer. Pathway analysis was used to understand the molecular pathogenesis of AD.

Regarding the (bio)sensors-based data, there is a plethora of such data for AD, enhancing the need for more DL and ML processes to capture their challenges. In this direction, this study [73] explores the use of wireless body sensor networks (WBSNs) and DL algorithms for diagnosing AD at an early stage. WBSNs are a novel technology consisting of multiple sensors implanted in or on the human body to track various physiological parameters, such as temperature, blood pressure, ECG, and EEG. Computer-aided algorithms have shown significant promise in scientific research, but no practical diagnostic approach is currently available for AD. As a result, there is a growing interest in applying DL to medical diagnoses, including AD.

Furthermore, a promising approach to early AD detection using wearable bio-sensor devices, potentially enabling early interventions and improving patient outcomes, was shown [74]. Herein, the collected data from wearable bio-sensor devices was analyzed using un-supervised machine learning to separate participants into distinct phenotypic groups based on their bio-metric data. The un-supervised machine learning approach involved a clustering process that grouped participants with similar bio-metric data into clusters or phenotypic groups. The clustering process helped to identify distinct patterns in the bio-metric data and identify sub-groups of participants that may be at higher risk for AD. The authors showed that data clustering is a crucial aspect of identifying meaningful patterns, enabling early interventions, and improving patient outcomes.

Beyond the bio-sensors, visual attention, as a typical eye-tracking behavior, is of great clinical value in detecting cognitive abnormalities in AD patients. The proposed multi-layered comparison convolution neural network (MC-CNN) showcases the contribution of DL techniques in diagnosing AD based on eye-tracking behaviors [75]. It has the potential to distinguish the differences in visual attention between AD patients and normal individuals. The MC-CNN utilizes hierarchical convolution to obtain multi-layered representations of heatmaps, which are then integrated into a distance vector to benefit the comprehensive visual task. Additionally, DeTrAs is a comprehensive study under the sensors-based perspective, proposing a DL-based Internet of Health framework for the assistance of Alzheimer patients [76]. It comprises three phases: a recurrent neural network-based Alzheimer prediction scheme that uses sensory movement data, an ensemble approach for abnormality tracking that includes a convolutional neural network-based emotion detection scheme and a timestamp window-based natural language processing scheme, and an IoT-based assistance mechanism for AD patients. DeTrAs highlights the potential of advanced sensors and DL techniques to provide personalized assistance to severely mentally affected patients, such as AD patients, in a patient-centric healthcare system. Furthermore, the development of a hybrid CNN architecture that leverages the strengths of multiple CNN models (Darknet53, InceptionV3, and Resnet101) for Alzheimer’s disease brain MRI classification was proposed [77]. The approach also utilizes the mRMR feature selection method to optimize the extracted features and improve the classification performance. The study demonstrates the potential of using hybrid CNN architectures and feature selection techniques in medical imaging applications for disease diagnosis. The use of multiple CNN models and feature selection can help capture more diverse and complementary information from the input images, leading to improved classification accuracy. Table 2 summarizes cutting-edge computational approaches to non-invasive sensor-based AD data.

## 5. The Role of AI and DL as Novel Insights in AD Monitoring

While non-invasive methods hold potential for early prediction of AD, the data generated from these techniques poses several computational challenges that need to be addressed. The data generated by such procedures can be extremely complex in terms of noise, volume, dimensionality, and heterogeneity. For instance, analyzing thousands of MRI or PET images presents a computational complexity that cannot be managed by simple methods since the presence of noise in these images makes their analysis difficult [78]. Similarly, data from blood screening often comprises omics (genomics, transcriptomics, and proteomics) measurements with an ultra-high number of samples and features [79]. A typical transcriptomics data consists of thousands to tens of thousands of samples (cells or tissues) and up to around 20,000 dimensions (genes). In addition, data obtained from wearable devices or bio-sensors is voluminous and continuous, and hence necessitates specialized methods for effective analysis. 

This complexity makes it challenging to analyze and interpret using traditional statistical methods. Therefore, the use of AI and deep learning (DL) algorithms has become increasingly popular in recent years, as they are well-suited to handle such large and complex datasets [80]. AI and DL techniques can identify patterns and relationships in data that may be overlooked by other analysis methods, and can generate predictive models that can assist in the early detection and management of AD. These techniques can also be used to develop personalized risk profiles for individuals based on their unique combination of risk factors, which can facilitate early interventions and improve outcomes. Therefore, the integration of non-invasive data with AI and DL has the potential to revolutionize the field of AD prediction and management. 

There are numerous benefits to using these sophisticated computational approaches to derive potential non-invasive biomarkers. The accuracy of such frameworks is directly proportional to the amount of data available for analysis. As the amount of available data increases, the accuracy of AI and DL models also increases. This inherent attribute lies in the fact that these models are designed to learn from patterns and relationships within datasets, and the more data that is available for analysis, the more patterns and relationships that can be identified. This results in models that are more comprehensive and accurate in their identification and characterization of potential biomarkers. Meanwhile, we are currently living in a big data era where we have an unprecedented amount of available data generated by non-invasive techniques for the early detection of Alzheimer’s disease. Hence, all these non-invasive technologies (imaging, blood-based, bio-sensors, wearable devices, etc.) enable the collection of large and complex datasets that can be used to identify potential biomarkers of early AD detection [81]. The benefits of these techniques in this context are significant and have the potential to revolutionize the way we can diagnose AD in its early stages, leading to improved outcomes for AD patients and a more efficient healthcare system (Figure 1). 

### 5.1. The Potential of Explainable AI Perspective

Although researchers have focused their attention on developing AI systems that can contribute to the early detection and prediction of AD, the lack of interpretability and transparency in these AI models has limited their effectiveness in clinical applications. This has led to a growing interest in the development of explainable AI (xAI) approaches that can provide insights into the decision-making processes of these models. It is a branch of AI that focuses on developing AI systems that can provide clear and understandable explanations of their decision-making processes to humans. It creates models that are transparent, and interpretable, and can explain their reasoning and actions in a way that humans can understand. This is important because traditional AI models can be complex and challenging to interpret, which can limit their trustworthiness and effectiveness. By making AI systems more explainable, xAI can help to increase human trust in these systems, improve the quality of decision-making, and enable human oversight and intervention where necessary. Hence, the development of xAI approaches that leverage data from non-invasive techniques has the potential to revolutionize the field of AD prediction [82]. This potential lies in the nature of xAI models, which will provide greater insight into the underlying mechanisms of AD. 

The most indicative xAI method for non-invasive data is the feature selection process, which identifies and selects the most important features or variables from a larger set of features that are used to train a model or to discriminate various classes (e.g., health vs. AD) [83]. It follows the principles of xAI by improving the transparency and interpretability of machine learning models. Furthermore, given that we have available an avalanche of non-invasive data, feature selection can help to simplify the model and make it more interpretable. There are various techniques used in feature selection, such as filter methods, wrapper methods, and embedded methods. In filter methods, features are ranked based on their relevance to the target variable, using statistical tests or other measures of correlation or mutual information. Features with low scores are then discarded, while the most important features are retained for model training. Wrapper methods, on the other hand, involve training a machine learning model on different subsets of features and selecting the sub-set that produces the best performance. This method is more computationally expensive than filter methods but can provide better accuracy in selecting the most relevant features. Embedded methods involve feature selection as part of the model training process. For example, some machine learning algorithms, such as Lasso and Ridge regression, can automatically select the most important features during model training. 

An indicative study [84] exploits the xAI nature of ML techniques by identifying the most dominant features in a big dataset. It combines three different feature selection methods using a ranking strategy to identify the best genes for classification tasks, with an emphasis on using tree-based ML techniques to achieve explainability and interpretability of the results. This approach provides a useful tool for the analysis and interpretation of single-cell RNA-sequencing data and may facilitate the early diagnosis of Alzheimer’s disease.

Summarizing, feature selection is a useful xAI method that can help improve the interpretability and transparency of machine learning models. By selecting the most relevant features, feature selection can help to simplify models, reduce overfitting, and identify potential biases, making it easier for humans to understand and trust the model’s decision-making processes [85]. 

### 5.2. The Potential of Deep Learning Perspective 

DL is a powerful technique for image analysis that has revolutionized the field of computer vision and has shown promise in predicting the onset of AD based on various bio-medical images such as MRI and PET. Image analysis involves the use of algorithms to extract meaningful information from digital images, and deep learning has emerged as one of the most effective methods for achieving this goal [86]. DL is under the AI umbrella, and it is a sub-set of machine learning that involves the use of neural networks, which are designed to simulate the function of the human brain. DL models consist of multiple layers of inter-connected nodes, or neurons, which are trained on large datasets of images to learn patterns and features that are useful for image analysis. 

DL thrives on data, and the more data it has access to, the more precise and accurate its outcomes become. In the field of AD prediction, there is a wealth of bio-medical images available for analysis, including MRI scans, PET scans, and other imaging modalities. By applying DL to these large datasets, we can hypothesize that it may be possible to identify key patterns and features that are predictive of AD onset. 

xAI methods can potentially help to improve outcomes in AD prediction using DL models that show promise in accurately predicting AD onset; however, their black box nature makes it challenging to understand how they rise to their predictions. By incorporating xAI methods into DL models for AD prediction, we can better understand the features and patterns that the model is relying on to make its predictions [87]. This can help to identify which features are most predictive of AD onset and provide greater transparency and interpretability of the model’s decision-making process. 

For example, DL models that use xAI methods to analyze imaging data, such as MRI scans, can help identify which regions of the brain are most affected by AD and provide insights into the progression of the disease. This information can be used to develop personalized treatment plans and improve patient outcomes [88]. Hence, incorporating xAI methods into DL models for AD prediction allows for improved outcomes by providing greater transparency and interpretability, identifying key predictive features, and enabling personalized treatment plans. 

## 6. Conclusions

In view of the ever-increasing volumes of bio-medical data obtained through non-invasive techniques, a new era of medicine is rising, with digitalization offering significant advantages in terms of reduced costs, patient discomfort, and risk of side effects. With the emergence of xAI and DL techniques, non-invasive-driven data could play an important role in predicting AD, transforming the perspective of medicine. AD diagnosis is of utmost importance since it could offer a better quality of life for patients by taking preventative measures. For these purposes, there is a growing demand for the identification of digital biomarkers that could detect early-stage deviations from normal cognition to AD. As non-invasive technologies advance and new data is generated, the active participation and contribution of AI and DL will be enhanced and continuously improved. By implementing these computational tools alongside the non-invasive bio-marker approach, promising results in the early diagnosis of AD could be achieved. 

## Figures and Tables

**Figure 1 sensors-23-04184-f001:**
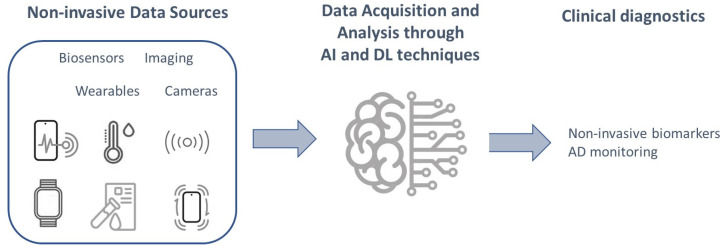
Illustration of AI and DL applications in AD using non-invasive data for clinical biomarkers identification.

**Table 1 sensors-23-04184-t001:** Non-invasive approaches to AD monitoring presented in the current study.

Non-Invasive Approach	Targeting/Monitoring	Ref.
OpenSMILE v2.1 toolkit	Evaluation of acoustic features	[21]
Immunoassay	p-tau181 detection	[25]
Non-Faradaic platform, metallization between the protein and ferric ion in redox probe	A*b*1-42 measurement in human serum	[29]
Electrochemical aptasensor	A*b* detection	[30]
Gold electrode with gold dendrite and dendritic electropolymerized poly(pyrrole-3-carboxylic acid substrate)	immobilization of prion protein and the selective detection of A*b* oligomers	[31]
Label-free electrochemical bio-sensor including thiol-terminated ssDNA aptamer receptors, attached to gold electrodes	A*b* oligomers recognition	[32]
Electrochemical immunosensor platform using gold functionalized nanoparticle	detection of A*b* peptides	[33]
Graphene oxide-based field-effect transistor bio-sensor	acetylcholinesterase and acetylcholine monitoring	[34]
Graphene oxide based single-use electrochemical Bio-sensor	detection of serum miRNA-34a	[35]
Screen-printed carbon electrode modified with PBA acid NHS ester	electrochemical detection of clusterin	[36]
Amperometric immunosensor/screen-printed carbon electrodes	tau measurement by implementing a sandwich immunoassay	[37]
Neutral charged immunosensor	Tau measurement	[38]
Bio-sensing platform consisting of indium tin gold electrode coated by PET	tau-441 measurement in serum	[39]
Electrochemical aptamer-antibody sandwich assay	tau-381 measurement in serum	[40]
F-FDG PET brain scans	a 48-layer deep convolutional neural network training	[43]
MRI images with a pre-trained VGG convolutional neural network	distinguish between AD patients and normal controls, EMCI and LMCI	[44]
PETNet, a graph-based convolutional neural network architecture	PET scan analysis	[45]
Eye-hand tasks recorded with a head-mounted video infrared eye-tracking system	visuomotor network dysfunctions	[47]
Eye-tracking tests	cognitive functions discrimination between controls, MCI, and AD patients	[48]
System including smartphones, tablet, eye trackers, microphone array wristband	AD cognitive assessment through record data	[50]
Low-cost robotic interface	oculomotor functions record in AD patients	[51]
Eye-tracking glasses	head tremor and eye blink	[52]
Smartphone sensors	AD mobility assessment	[53]
Smart terminal device for screening finger function	records finger dexterity to facilitate the screening of MCI and AD	[54]
System for body signals monitoring (heart rate and skin temperature) and motion location tracking	abnormal behavior in daily motion and gait abnormalities	[55]
Foot-mounted wearable sensor-based device	correlation of aerobic activity along with traditional cognitive protocols	[56]
Wrist-worn wearable accelerometer	sedentary behavior and bout	[57]
Heart rate sensors using PPG technology	monitoring heart rates in AD patients	[61]
Bio-sensing platform comprising a wristwatch, a wireless pulse oximeter, a PPG and gait sensors	discriminating symptoms of MCI and cognitively healthy subjects	[63]

**Table 2 sensors-23-04184-t002:** Computational methods applied to non-invasive AD data presented in the current study.

Origin of Non-Invasive Data	Technique	Computational Method	Ref
Wearable sensors	Gait Data	Elimination method-based ensemble and oversampling model	[64]
Wearable IoT devices	Speech data	Multiple ML models	[65]
Imaging	MRI	Convolutional Neural Network (3D CNN)	[66]
Imaging	MRI	CNN and SVM	[67]
Imaging	MRI	Combination of Gaussian Mixture Model (GMM), CNN for image segmentation, combination of Extreme Gradient Boosting (XGBoost) and SVM for classification	[68]
Imaging	Neutrophil images	Deep Learning	[71]
Imaging	MRI	Convolutional Neural Network	[77]
Blood	Transcriptomics	Machine Learning	[69]
Blood	small non-coding RNAs	Various machine learning approaches (support vector machines, decision trees, neural networks, gradient boosted trees)	[70]
Blood	Circulating cfDNA (Methylation)	Deep Learning, SVM, Generalized Linear Model (GLM), Prediction Analysis for Microarrays (PAM), Random Forest (RF), and Linear Discriminant Analysis (LDA)	[72]
Bio-sensors	Wireless Body Sensor Networks	Deep Learning algorithms for AD diagnosis	[73]
Bio-sensors	Wearable bio-sensor device data	Un-supervised machine learning (Clustering)	[74]
Sensors	Eye-Tracking	Deep Learning	[75]
Sensors	Sensory movement data	Deep Learning	[76]

## Data Availability

Not applicable.

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
