# Peer review of "Revolutionizing the Early Detection of Alzheimer’s Disease through Non-Invasive Biomarkers: The Role of Artificial Intelligence and Deep Learning"

_sensors, 2023, doi:10.3390/s23094184_

Round 1

Reviewer 1 Report

In their submission, the authors provide a perspective on the use of artificial intelligence (AI) and deep learning in early diagnosis of Alzheimer's disease. The topic of the submission is timely, with a tremendous interest being placed currently in early diagnosis of neurodegenerative diseases and the applicability of AI and big data approaches. 

As a general comment, in its current form, the manuscript places somewhere in between an opinion/perspective paper and a review paper, with a slight tendency towards the former. I believe that the authors should clarify this as early in their manuscript as possible (abstract or introduction), in order to not mislead the readership. If they opt towards the review, then they definitely must provide more details on current state-of-the-art, particularly on multimodal diagnosis of AD using deep learning, which is a burgeoning area with a large number of recent manuscript. And, preferably, more technical details on existing approaches.

In case they steer towards an opinion/perspective paper, then they must strengthen their case with deeper insight into e.g. what technologies are more likely to achieve impact, what should be expected in the near future.

In either case, it would be beneficial for the manuscript to include more references on current state-of-the-art (e.g. in sections 4.1 and 4.2) and the addition of one or more tables summarizing the details provided in the text would help.   

Specific points:

- Section 4.1 Explainable AI: this section is too succinct and presents too generically the xAI techniques. More details related to AD diagnosis would be appreciated, including more relevant references.

- Similarly section 4.2 (Potential of Deep Learning Perspective) is to narrative in its current form. Only 2 references are provided, which is too few, given the vast amount of recent literature in this area.

Author Response

Reply to Comments and Suggestions

In their submission, the authors provide a perspective on the use of artificial intelligence (AI) and deep learning in early diagnosis of Alzheimer's disease. The topic of the submission is timely, with a tremendous interest being placed currently in early diagnosis of neurodegenerative diseases and the applicability of AI and big data approaches. 

As a general comment, in its current form, the manuscript places somewhere in between an opinion/perspective paper and a review paper, with a slight tendency towards the former. I believe that the authors should clarify this as early in their manuscript as possible (abstract or introduction), in order to not mislead the readership. If they opt towards the review, then they definitely must provide more details on current state-of-the-art, particularly on multimodal diagnosis of AD using deep learning, which is a burgeoning area with a large number of recent manuscript. And, preferably, more technical details on existing approaches.

In case they steer towards an opinion/perspective paper, then they must strengthen their case with deeper insight into e.g. what technologies are more likely to achieve impact, what should be expected in the near future.

Reply: Thank you for your comment. We stated in the abstract that this work attempts to examine some of the facts and the current situation of non-invasive approaches to AD diagnosis leveraging the potential of AI and DL. In this direction, we have strengthened the manuscript by including more recent studies that examine these issues in detail. Although it is not an opinion paper, we were willing to stress the importance of non-invasive methods and AI and DL in handling non-invasive data for the diagnosis of AD. For this purpose, we added more references accordingly.

In either case, it would be beneficial for the manuscript to include more references on current state-of-the-art (e.g. in sections 4.1 and 4.2) and the addition of one or more tables summarizing the details provided in the text would help.   

Reply: We included more than 40 references in all sections, we improved the previous Section 4 adding also a new one (now reads section 4 and section 5). Also new Tables 1 and 2 are included according to your comment.

Specific points:

- Section 4.1 Explainable AI: this section is too succinct and presents too generically the xAI techniques. More details related to AD diagnosis would be appreciated, including more relevant references.

- Similarly section 4.2 (Potential of Deep Learning Perspective) is to narrative in its current form. Only 2 references are provided, which is too few, given the vast amount of recent literature in this area.

Reply: A new section 4 describes relevant studies which present AI and DL applications related to this field. Thank you again for your constructive suggestion which help manuscript improvement.

Reviewer 2 Report

I reviewed the study titled "Revolutionizing the early detection of Alzheimer's Disease through Noninvasive Biomarkers: The role of Artificial Intelligence and Deep Learning" in detail. The missing points in the study are presented as items.

Researchers stated that artificial intelligence and deep learning-based models can be used for the diagnosis of AD. These methods are not included in the abstract. The proposed method is not mentioned. No outcome value was observed in the paper. However, there are many studies on Alzheimer's disease in the literature. Tables and figures were not used in the study. Tables and figures are very important for understanding the study. The organization of the article should be reviewed. I don't think the content is appropriate. There are many artificial intelligence and deep learning methods in the literature. It is necessary to categorically examine the studies on these and discuss the results. The datasets used in these studies should be included in this discussion. It is essential to reorganize the article for this and more shortcomings. In summary, I did not understand what the purpose of the paper was.

Author Response

Reply to Comments and Suggestions

Thank you for your critical comments. We added new sections and tables including recent studies (more than 40) and clarify some misunderstandings in whole manuscript according to your suggestions. Focusing on AL and DL a new section 4 is presented. According to your last comment regarding the datasets used in these studies,  we should state we did not proceed an analysis as this is a review paper and anyone has access to the referenced articles. We believe this point is clear. We hope the revised manuscript could address your concerns and thank you for your positive suggestions.

Reviewer 3 Report

In this short review, Vrahatis et al describe about noninvasive technique to detect Alzheimer’s Disease (AD) in early stage. It is true that AD is a silent pandemic and due to its high increasing rate significant effort has been paid to develop efficient detection technique and cure process. But still, we are so far from our goal. In this respect this review provides a general idea in this area. In my opinion this review is not only useful for scientist deal with AD but also the people out of this area. So, I recommended to accept this review after following changes.

1. The presentation throughout the manuscript is very nice, but in the 'reference' section author use doi link for some reference (eg. 12, 13 etc). As this reference have "article name, year, volume, page number", so no need the doi link. I recommend removing the doi link.

Author Response

Reply to Comments and Suggestions

We would like to thank Reviewer for his/her positive comments. Based also on his/her clear opinion we added more recent studies highlighting the importance of non-invasive techniques in clinical practice.

Reviewer 4 Report

This manuscript covers an interesting and essential topic, “Revolutionizing the early detection of Alzheimer’s Disease through Noninvasive Biomarkers: The role of Artificial Intelligence and Deep Learning”. Overall, the manuscript has some strengths, but also various weaknesses, as outlined below. I have suggestions below as to how the manuscript could be improved.

In introduction section, 2023 ALZHEIMER'S DISEASE FACTS has been released and author should update most recent facts for AD (PMID: 36918389).

Very limited number of studies included in this review and large number of important studies are missing here (e.g. PMID: 29955666, 35478701, 36856708, 35460969, 32603732, 35479927 etc..), and the studies mentioned are also not exhaustive, as only 3-4 studies are mentioned here in each section. Please include at least 10-15 more relevant and original studies in each section of Noninvasive techniques to make it more elaborative and advance.  

Again, many original important studies are missing in “role of Artificial Intelligence and Deep Learning” section like (PMID: 35681440, 33045895, 33665339, 34322704), like above please include at least 10-15 more relevant original studies to elaborate this section appropriately.

All the reported non-invasive techniques sections do not appear to depict the entirety of the literature and thus it appears like the authors selected particular studies and omitted others to suit their narrative. The discussion should instead be more nuanced and include studies reporting negative or divergent results as well.

It appreciates that Author summarizes the various techniques including wearable devices, sensor-based biomarkers etc., with its advantages and limitations in tabular form. Its easy for the readers.

Author Response

Reply to Comments and Suggestions (point-by-point)

This manuscript covers an interesting and essential topic, “Revolutionizing the early detection of Alzheimer’s Disease through Noninvasive Biomarkers: The role of Artificial Intelligence and Deep Learning”. Overall, the manuscript has some strengths, but also various weaknesses, as outlined below. I have suggestions below as to how the manuscript could be improved.

In introduction section, 2023 ALZHEIMER'S DISEASE FACTS has been released and author should update most recent facts for AD (PMID: 36918389).

Reply: Thank you for your comment. We added the suggested reference as well as some further along these lines.

Very limited number of studies included in this review and large number of important studies are missing here (e.g. PMID: 29955666, 35478701, 36856708, 35460969, 32603732, 35479927 etc..), and the studies mentioned are also not exhaustive, as only 3-4 studies are mentioned here in each section. Please include at least 10-15 more relevant and original studies in each section of Noninvasive techniques to make it more elaborative and advance. 

Reply: Recommended studies and additional ones (approx. 25 new studies) were included in the noninvasive techniques section.

Again, many original important studies are missing in “role of Artificial Intelligence and Deep Learning” section like (PMID: 35681440, 33045895, 33665339, 34322704), like above please include at least 10-15 more relevant original studies to elaborate this section appropriately.

Reply: We included a new Section 4 (now reads "Recent Advances in Noninvasive AD Diagnosis using DL and ML Techniques") and a new Table 2 according to your comment.

All the reported non-invasive techniques sections do not appear to depict the entirety of the literature and thus it appears like the authors selected particular studies and omitted others to suit their narrative. The discussion should instead be more nuanced and include studies reporting negative or divergent results as well.

Reply: Thank you for your comment. We included more studies according to your suggestion (please see the revised Section 3.4). Furthermore, we added three new sections (please see new 3.6, 3.7 and.8) focused on oculomotor and movement functions as well as speech and language functions and autonomic nervous system functions.

It appreciates that Author summarizes the various techniques including wearable devices, sensor-based biomarkers etc., with its advantages and limitations in tabular form. Its easy for the readers.

Reply: Thank you for your suggestion. We included a new Table 1 which summarized the various techniques.

We would like to thank the Reviewer again for his/her constructive comments that help to improve the manuscript.

Round 2

Reviewer 2 Report

After the revision, I reviewed the study titled "Revolutionizing the Early Diagnosis of Alzheimer's Disease Through Noninvasive Biomarkers: The Role of Artificial Intelligence and Deep Learning" in detail. After the revision, the article has gotten to a much better point. Congratulations to the researchers. The work now needs minor changes. The summary part has been edited. Titles 3.6, 3.7 and 3.8 were added later and contributed a lot to the article. The newly added Table 1 is definitely an essential part of the article. It is possible to expand the corresponding table. A new paragraph can be added at the end of the first section about the organization of the article and the contribution of the study to the literature. "https://doi.org/10.1002/ima.22632" I recommend you to examine the related studies and expand Table 1,2. Having at least one shape in the article is important for visuality.

Author Response

Thank you for your kind words regarding our revised manuscript and your constructive last comments. We proceeded accordingly. A new paragraph was added at the end of the first section about the organization of the article. We expanded tables and the corresponding sections adding more references. We also discussed the suggested article and we included a new Figure 1 following your recommendation.

Thank you again for your positive suggestions.

Reviewer 4 Report

This manuscript can be published in current format

Author Response

We would like to thank the Reviewer for his positive suggestion.